# Human Gingival Fibroblast and Osteoblast Behavior on Groove-Milled Zirconia Implant Surfaces

**DOI:** 10.3390/ma15072481

**Published:** 2022-03-28

**Authors:** Mariana Brito da Cruz, Joana Faria Marques, Neusa Silva, Sara Madeira, Óscar Carvalho, Filipe Samuel Silva, João Caramês, António Mata

**Affiliations:** 1Universidade de Lisboa, Faculdade de Medicina Dentária, Unidade de Investigação em Ciências Orais e Biomédicas (UICOB), LIBPhys-FTC UID/FIS/04559/2013, Rua Professora Teresa Ambrósio, 1600-277 Lisboa, Portugal; jmarques2@campus.ul.pt (J.F.M.); carames@campus.ul.pt (J.C.); admata2@campus.ul.pt (A.M.); 2Universidade de Lisboa, Faculdade de Medicina Dentária, Unidade de Investigação em Ciências Orais e Biomédicas (UICOB), Rua Professora Teresa Ambrósio, 1600-277 Lisboa, Portugal; neusa.silva@edu.ulisboa.pt; 3Center for Microelectromechanical Systems (CMEMS), Department of Mechanical Engineering, University of Minho, 4800-058 Guimarães, Portugal; saramadeira@dem.minho.pt (S.M.); oscar.carvalho@dem.minho.pt (Ó.C.); fsamuel@dem.uminho.pt (F.S.S.); 4Universidade de Lisboa, Faculdade de Medicina Dentária, Bone Physiology Research Group, Rua Professora Teresa Ambrósio, 1600-277 Lisboa, Portugal; 5Universidade de Lisboa, Faculdade de Medicina Dentária, Cochrane Portugal, Instituto de Saúde Baseada na Evidência (ISBE), Avenida Professor Egas Moniz, 1649-028 Lisboa, Portugal

**Keywords:** zirconia, dental implants, milling, osteoblasts, fibroblasts

## Abstract

Two type of cells representing periodontal hard tissues (osteoblasts) and soft tissues (fibroblasts) were evaluated in response to microgroove-milled zirconia surfaces. A total of 90 zirconia discs were randomly assigned to four width-standardized milling microgroove-textured groups and a control group without grooves (UT). The sandblast and acid-etch protocol were applied to all samples. Both cell lines were cultured on zirconia discs from 1 day up to 14 days. Cell morphology and adhesion were evaluated after 1 day of culturing. Cell viability and proliferation of the cells were measured. Alkaline phosphatase activity, collagen I, osteopontin, interleukin 1β and interleukin 8 secretions were assessed at predefined times. The results obtained were presented in the form of bar graphs as means and standard deviations. Multi comparisons between groups were evaluated using two-away ANOVA or Mann–Whitney tests, and a *p*-value < 0.05 was established. Group comparisons with regard to cell viability, proliferation and secretion of collagen I, interleukin-1β and interleukin 8 revealed no statistically significant differences. The alkaline phosphatase activity and osteopontin secretion were significantly higher in the group with a large groove compared to the small one and the control group. Nevertheless, the viability of gingival and bone cells did not appear to be affected by the milled microgroove texture compared to the conventional sandblasted and acid-etched texture, but they seem to influence osteoblasts’ cellular differentiation.

## 1. Introduction

Since their discovery, dental implants have revolutionized implantology and have become a routine treatment in everyday dental practice. Despite the high survival rate of implant treatment [1], inflammatory conditions around osteointegrated dental implant have been reported resulting in bone and soft tissue losses [2,3,4,5]. The frequent occurrence of these phenomena in daily oral rehabilitation underscores the need for new materials and improved implant surfaces that can promote osseointegration and preserve bone and gum tissue around dental implants.

The first material successfully used in dentistry for dental implants was titanium, but the drawbacks identified over the years, such as the release of nanoparticles, allergic reactions and poor aesthetic associated with a gray color, led to the search for new materials such as ceramics [1,6,7,8]. The recent technology advancement in dental materials led to an increase in the number of commercially available ceramic materials for clinical use, such as alumina ceramic, machinable glass ceramic, feldspathic porcelain, zirconia ceramic and others. Among them, yttria-stabilized polycrystalline tetragonal zirconia (YTZP) is the most popular because of its mechanical properties and similarity to titanium. In addition, due to its metal-free properties, coupled with its aesthetic white color and translucency like human teeth, YTZP could be the preferred material for dental implants in the future as an alternative to titanium implants [9,10,11,12,13].

The effectiveness of dental implants is mainly based on the bone-implant interaction [14]. However, several factors can influence this process such as the: chemical composition, surface wettability, roughness and topography of the dental implant surface [15,16,17].

Surface modifications of dental implants such as the one mentioned above, surface roughness, are often used on implants made of zirconia, as it has been recognized that they affect the in vitro cell response and implant integration into the bone in vivo. 

These micro irregularities conceived by sandblasting and acid etching (SBAE) improve the biocompatibility of the implants, increasing contact surface area and adhesion of osteogenic cells. The microroughness from 1 to 100 μm it is obtained by blasting and acid etching resulting in the improvement of osteoblast biological behavior [13,18,19,20]. 

Therefore, the topography of the implant surface can be divided into three levels depending on its characteristics: macro (10 μm to 1 mm), micro (1–10 μm) and nano (1–100 nm) scale [15]. These characteristics have been mentioned in recent studies of titanium implants as having a significant impact on promoting bone ingrowth [21], but this has hardly been investigated in the case of YTZP dental implants. Overall, it has been described that microtopography is a critical determinant of human cells’ adaptation and differentiation [22], due to the production of directional physical signals in cell regulation and in the assembly and collagen matrix orientation [23].

In that regard, the standardization of implant surfaces by performing topographies on them has been described over the years [24,25,26]. Among the various types of topographies designed, microgroove topographies have been widely explored for their impact on cell alignment, as they can be generated relatively conveniently using a range of microfabrication techniques such as conventional milling. In vitro studies of osteogenic cell behavior showed that they are strongly oriented towards the direction of the grooves when compared to surfaces without texture, on which arbitrary alignment is usually noticed. However, the ideal texture and dimensions for optimal osteointegration are still being extensively researched, and most of these studies are performed on titanium implant surfaces. It is therefore important to know whether creating microgroove patterns on implants made of zirconia would be beneficial for hard and soft tissues [22,26].

Thus, the purpose of this investigation was therefore to understand whether cell behavior of osteoblasts and fibroblasts on width-standardized milling microgroove-textured zirconia surfaces is improved compared to sandblasted and acid-etched microtopography.

## 2. Materials and Methods

### 2.1. Substrates 

Ninety zirconia discs with 8 mm in diameter and 2 mm in thickness made of a commercially available 3 mol % yttria-stabilized zirconia powder (3Y-TZP) (3YSB-E Tosoh Corporation, Tokyo, Japan) with 40 nm in particle size and 60 μm in average cluster size. Their chemical compounds are described in Table 1. 

The production was based on the pressing and sintering technique. After performing the pressing process (210 MPa) in a steel mold that has 10 nm of internal diameter and 50 nm in height, all samples except for the control group (UT) were textured with width-standardized milling microgroove defined as groups A, B, C and D, according to Table 2.

Once the surfaces were textured, all samples were sintered in an oven (Zirkonofen 700, South Tyrol, Italy) and then cleaned with isopropyl alcohol using ultrasonic equipment. The samples were then sandblasted for 30 s with particles of alumina (Al_2_O_3_) with an average size of 250 μm, 6 bars of pressure and 12 cm of distance, and washed again with ultrasonic equipment and isopropyl alcohol. After this step, each sample was immersed in hydrofluoric acid (48% HF) at room temperature for half an hour and cleaned once more with ultrasonic equipment, where it was immersed in isopropyl alcohol for 5 min, resulting in a surface roughness of 1.45 μm. In the final step, an ultrasonic bath process was once again performed for all the samples with absolute ethanol and sterilized in the autoclave in order to carry out all biological tests (Figure 1). All the biological tests were repeated in three independents experiments.

The final aspect of the samples from each group after the surface treatment was observed and evaluated by SEM JSM-6010 LV (JEOL Ltd., Tokyo, Japan). SEM JSM-6010 LV images were obtained at 500× magnification at 10 kV acceleration voltage. Backscattering Electron Detector (BSED) images were taken at 15 kV, (Figure 2). FEG-SEM images reveal similar topographies in all textured samples.

### 2.2. Cell Culture

Human Gingival Fibroblasts—HGFs (HGF; Applied Biological Materials Inc., Richmond, BC, Canada) were cultured in a supplemented Dulbecco’s Modified Eagle’s Medium—DMEM (Lonza^®^, Basel, Switzerland) with 10% Bovine Fetal Serum (Biowest^®^, Nuaillé, France) and 1% Penicillin with streptomycin (G255 Applied Biological Materials Inc., Richmond, BC, Canada).

Human Fetal Osteoblasts—hFOBs 1.19 (CRL-11372^TM^; American Culture Collection, Manassas-ATCC^®^, Manassas, VA, USA) were cultured in a mixture (1:1 *v*/*v*) of DMEM (Dulbecco’s modified Eagle’s Medium-DMEM) from (BiowhittakerTM, LonzaTM, Basel, Switzerland) and Ham’s F-12 Medium (Sigma-Aldrich^®^ 51651C, St. Louis, MO, USA), in which was added 0.3 mg·mL^−1^ of G418 (InvivoGgen, Toulouse, France) and 10% of Bovine Fetal Serum (Biowest^®^, Nuaillé, France).

Both cell lines were incubated at 37 °C with 5% CO_2_ and 98% humidity. When the cells reached approximately 80% confluence, trypsin-EDTA (Lonza, Veners, Belgium) was added to detach them, they were later centrifuged, and the pellet was resuspended in the respective medium. To perform each cell culture assay 1 × 10^4^ cells/mL cells at a 4th passage were seeded in 48-well plates containing sterile sample (Corning, NY, USA). 

#### 2.2.1. Fibroblast and Osteoblast Cells Viability and Proliferation 

The viability and proliferation of both cells was assessed with—Cell-Titer Blue^®^ reagent (Promega, Madison, WI, USA) following the manufacturer’s instructions. After 1, 3, 7 and 14 days of the cells culturing, fluorescence intensity was measured in arbitrary fluorescence units (AU). The detection range used was excitation/emission wavelengths of 560/590 nm, measured with a luminescence spectrometer (PerkinElmer LS 50B, Waltham, MA, USA). N = 15 samples cultured with osteoblasts and fibroblasts were analyzed per group for the viability and proliferation of the two cell lines.

#### 2.2.2. Fibroblasts and Osteoblasts Morphology

To determine the cell morphology, samples with osteoblasts and fibroblasts were observed after one day of culturing. After washing, all cell samples were fixed with 1.5% glutaraldehyde and dehydrated with increasing ethanol concentrations (70%, 80%, 90% and 100%). The samples were incubated in Hexamethyldisilazane—HMDS (440191 Aldrich Chemistry, Milwaukee, WI, USA) and then covered with gold by the sputtering method (LEICA EM ACE600, Heerbrugg, Switzerland), with a 15 nm ultrathin gold-palladium film 80–20% in weight using high-resolution sputter coater (208HR Cressington Company, Watford, UK) coupled with an MTM-20 Cressington High Resolution Thickness Controller. Scanning Electron Microscopy—SEM JSM-6010 LV (JEOL Ltd., Akishima, Japan) was carried out at different magnifications (500, 2000, 5000×) at 10 Kv. Backscattering Electron Detector (BSED) images were taken at 15 kV. Two researchers performed the observations within the focus on morphology, spreading and early cell contact establishment with the materials. N = 3 samples cultured with osteoblasts and fibroblasts were analyzed for SEM images.

#### 2.2.3. The Activity of Alkaline Phosphatase (ALP) 

The activity of ALP was evaluated on osteoblasts at 7 days of culturing with fluorometric enzymatic tests from (ab83371 ALP Assay Fluorometric, Abcam, Cambridge, UK) following the instruction by the manufacturer. To evaluate the enzyme activity, a standard curve was elaborated. Both samples and standards were evaluated at excitation/emission wavelengths of 360/440 nm utilizing a Fluorescence spectrometer (PerkinElmer LS 45, Waltham, MA, USA) The ALP values were transformed into mU/µL according to the regression equation from the standard curve. N = 4 samples cultured with osteoblasts were analyzed per group for ALP activity.

#### 2.2.4. Quantification of Interleukin 1β (IL-1β) by ELISA Method

IL-1ß was quantified in osteoblast cell cultures after 3 days using an ELISA kit, the Human IL-1b Chemiluminescent (LumiAB^TM^, San Francisco, CA, USA), according to the manufacturer`s protocol. The samples were measured by a luminescence technique with a Victor Nivo Multimode Plate Reader (PerkinElmer^®^ Inc., Waltham, MA, USA). The results are obtained in absorbance units (nm) and converted into pg/mL by correlation with the linear regression equation obtained in the standard curve. To quantify IL-1β N = 4 were analized per group

#### 2.2.5. Quantification of Collagen I by ELISA Method

The quantification of collagen in each osteoblast and fibroblast cell culture was carried out after 7 days of culturing. HumanPro-Kit Collagen I alpha 1 DuoSet Elisa (DY6220 05 R&D Systems, Inc., Minneapolis, MN, USA) was used according to the manufacturer`s protocol. The fluorescence intensity of all the samples was detected at excitation/emission wavelengths of 540 nm using a Multimode Plate Reader (PerkinElmer^®^ Inc., Waltham, MA, USA). Results were acquired in units of absorbance (nm) and transformed into pg/mL according to the standard curve. N = 4 osteoblasts and fibroblasts with culture suspension were analyzed per group to quantify collagen I.

#### 2.2.6. Quantification of Osteopontin by ELISA Method

The quantification of osteopontin was carried out using the ELISA Chemiluminescent Human Osteopontin kit (LumiAB^TM^, San Francisco, CA, USA) using a luminescence technique with Multimode Plate Reader (PerkinElmer^®^ Inc., Waltham, MA, USA ) after 3 days of osteoblast culturing. The fluorescence intensity of the samples was detected at excitation/emission wavelengths of 700 nm. Results were acquired in units of absorbance (nm) and transformed into concentration (pg/mL) based on the standard curve. N = 4 osteoblasts with culture suspension were analyzed per group to quantify osteopontin.

#### 2.2.7. Quantification of Interleukin 8 by ELISA Method

Interleukin 8 was quantified with the Human IL-8 Chemiluminescent ELISA Kit (Lumi AB^TM^, San Francisco, CA, USA) using Multimode Plate Reader (PerkinElmer^®^ Inc., Waltham, MA, USA), at 1 day of fibroblast culturing. Results were acquired in units of absorbance (nm) and transformed into concentrations (pg/mL) based on the standard curve. N = 4 fibroblasts with culture suspension were analyzed per group to quantify IL-8.

### 2.3. Statistical Evaluation

The statistical evaluation was performed with IBM^®^ SPSS^®^ 24.0 for MacBook (SPSS, Chicago, IL, USA). The normality analyses were carried out for all data. Comparisons among groups in terms of viability, proliferation, ALP, interleukin 1β, collagen I, osteopontin and interleukin 8 were based on two-way ANOVA or Mann–Whitney. Tukey’s post-hoc analysis was used to find the statistical differences between the groups, and the significance level was stablished at *p*-value < 0.05. The obtained data were presented in terms of mean ± standard deviation (SD).

## 3. Results

### 3.1. Fibroblast and Osteoblast Viability and Proliferation 

The viability and proliferation results for the two cell lines, fibroblast and osteoblast, were attained for 1, 3, 7 and 14 days of culturing (Figure 3). 

The viability of osteoblasts increased over the 14 days of culturing, although without significant differences when comparing all groups (*p* > 0.05). Likewise, the rate of proliferation of the osteoblast cells showed greater cell growth in the first seven days of culturing. However, there are no statistically significant differences between any of the groups (*p* > 0.05). Regarding the fibroblasts’ behavior, the groups comparison revealed no statistically significant differences (*p* > 0.05). Over the culturing time, the cell proliferation rate was higher from 3 to 7 days of fibroblast culturing, but not statistically different between groups however (*p* > 0.05).

### 3.2. Fibroblast and Osteoblast Morphology 

Images obtained from FEG-SEM after 1 day of osteoblast culturing (Figure 4) showed adherent cells with similar morphologies in all groups, with the exception of the C group, in which the cells showed an elongated phenotype with larger cytoplasmatic extensions. The cell bodies appear to be distributed in the direction of the groove in all samples, and no differences in cell numbers were observed between any of the groups. In the culture with fibroblast cells (Figure 4), the images showed cell adhesion after 1 day of culturing. Fibroblasts showed a characteristic morphology in all groups, with many cell extensions and also flattened cell bodies According to the pattern of the samples, no cell distribution could be seen.

### 3.3. ALP Activity

ALP results based on the suspension of osteoblasts are shown in Figure 5. The results show the greatest ALP activity from the A group, but only with significant differences when compared to the B group (*p* < 0.05). When compared to all the study groups in regard to alkaline phosphatase activity, there were no statistically significant differences between all groups (*p* > 0.05).

### 3.4. Interleukin 1β

Figure 6 shows interleukin 1β production, and the results are comparable between the groups, with no statistically significant differences among all groups (*p* > 0.05).

### 3.5. Collagen Type I

The level of collagen type I in the extracellular medium was measured after 7 days in osteoblast and fibroblast cultures (Figure 7). The results were identical between all study groups and for both cultures, with no statistically significant differences between them (*p* > 0.05).

### 3.6. Osteopontin

The osteopontin concentration was assessed after 3 days of the osteoblast culturing (Figure 8). There was a marked increase in osteopontin production for all the samples, without statistically significant differences when all groups are compared (*p* > 0.05).

### 3.7. Interleukin 8

Fibroblast interleukin 8 production was analyzed after 1 day (Figure 9). No statistically significant differences were found after 1 day (*p* > 0.05). When the repeated measures were performed to compare the effect of all study groups on the concentration of interleukin 8, no statistically significant differences were found (*p* > 0.05). 

## 4. Discussion

Yttria-stabilized polycrystalline tetragonal zirconia is a promising alternative to titanium, as noted in current literature reviews, based on its biocompatibility, better soft tissue performance linked to aesthetic outcomes and similar tissue response to osteointegration [27,28,29,30]. In order to improve their biological behavior, several modifications are made to the surface of zirconia implants [5,31].

In this context, this in vitro study aimed to gain a deeper understanding of the effect of milled microgroove implants made of zirconia with width-standardized milling microgroove textures on the biological behavior of human fetal osteoblasts and human gingival fibroblasts compared to sandblasted and acid-etched surfaces, which are considered the gold standard surface modification for zirconia implant surfaces. 

Although a previous study was completed to find the drill bit diameter to be used during the texturing process and to understand the best parameters to obtain the desired dimensions, grooves with standardized widths and equal depths, the results for the topographic parameters analyzed turned out to be statistically different. Microgrooves made by conventional milling were observed with a width from 48 μm to 127 μm and a depth from 8 μm to 17 μm. However, these surface modifications associated with patterned surfaces from 10 μm to 1 mm have not been adequately studied on zirconia surfaces, and most in vitro studies contradict each other when it concerns the pattern dimensions [32,33,34].

In addition to the milled microgroove patterns created on tested groups, microtopography was performed on all groups by sandblasting and acid etching, a technique that makes it possible to discard the microroughness biases of surfaces, a parameter that has already been described as fundamental to the influence of cell adhesion. This is a limitation found in most of the studies that claim that the creation of patterns promotes the growth and cell adhesion of osteoblasts on zirconia surfaces without actually controlling specific parameters of the surfaces, such as surface roughness [35].

The evaluation of the adhesion and distribution of osteoblasts and fibroblasts on milled microgroove-patterned surfaces made of zirconia with standardized widths were carried out. The results showed a greater affinity of the osteoblasts for milled microgroove patterns as well as a more differentiated morphology of the cell bodies. Fibroblast cells showed no signs of changes in cell distribution according to the sample pattern in the studies, which was milled microgroove. The obtained results are in line with a previous study carried out by Zhu et al. in 2005 [36], in which they show that nano grooves play a central role in modulating osteoblast cell behavior and in the orientation and alignment of cell bodies. In addition, it is also described that osteoblasts on surfaces with grooves change their orientation and follow the groove direction [36]. A recent in vitro study carried out by Fernandes et al. demonstrates that osteoblast cells on zirconia implant surfaces texturized with microgrooves exhibited a characteristic filopodia and veil shape in the direction of the groove patterns created. Nonetheless, it was a study performed on zirconia implant surfaces with osteoblast cells, although the microgroove pattern was created with a Nd: YAG laser instead of conventional milling [26]. In addition to studies on dental material, a study by Sun et al. in 2016 on microgroove polystyrene surfaces with osteoblast cells found that, on unpatterned surfaces, they were randomly oriented, but on the microgroove surface, they were aligned following the grooves’ direction with marked and typical morphologies [37].

This study also shows that width-standardized milling microgroove textures do not appear to affect fibroblast and osteoblast viability and proliferation. The obtained results are in accordance with a study by Fernandes et al. [26] in which they examined the osteoblast cell on zirconia discs textured with a Nd:YAG laser, and they found an improvement in viability over time with no statistically significant differences compared to the control group. However, they report much higher viability values in the microgroove group compared to the control group. Another similar finding was made by Holthaus et al. from 2012 [38], in which the depth of the microchannel created by micro molding on ceramic surfaces had virtually no significant impact on osteoblasts, but the width of the groove seemed to determine the angle of growth of the osteoblasts. However, in the same study proliferation and collagen production were not influenced by the surface pattern created [38]. Different results were presented by Nadeem et al. [39], showing that human mesenchymal cells cultured in grooves larger than 50 μm showed a more favorable osteogenic response than grooves of 10 μm. 

After a detailed analysis of the results, abrupt differences in the surface roughness of the groups examined were found, and the authors themselves also point to a stronger cell alignment in the 10 μm grooves [39]. This cellular behavior can be explained by the fact that the micro roughness produced by SBAE is similar on all surfaces, with these microgrooves not adding any added value to the fibroblast and osteoblast cells’ behavior. The actual influence of the surface pattern on the biological behavior of osteoblasts and fibroblasts cannot be extrapolated despite the available in vitro studies [40,41].

While an obvious effect of width-standardized milling microgroove textures on the shape and orientation of osteoblast cells was observed, different results were found for their viability. However, some of the main markers of osteoblast differentiation appear to be influenced by width-standardized milling microgroove textures created on the zirconia implant surfaces examined in this study. It is therefore known that when an implant material is placed into an edentulous patient, an inflammatory process is triggered. This inflammatory response to the implant material has been linked to several cytokines that are important at the molecular level for the tissue regeneration process [42].

The expression levels of ALP, IL-1β, Collagen I, osteopontin and IL-8 on width-standardized milling microgroove textures were analyzed for osteoblast and fibroblast cell cultures. For human osteoblasts, ALP, IL-1β, Collagen I and Osteopontin were chosen as the main signal markers to assess the osteoblast phenotype, and no statistically significant differences were found in the expression levels of all width-standardized milling microgroove-textured groups compared to the control group, only with sandblasted and acid-etched group. After 7 days of osteoblast culturing, however, samples from the A group (127 μm in width) showed the greatest ALP activity, but only with significant differences compared to the B group (48 μm in width). ALP is known as a marker of bone formation and calcification, and it is present during bone mineralization [43]. Therefore, it appears that the width of 127 μm found on the A group favor early osteoblast growth and differentiation. In this sense, a study by Miyahara et al. on macroscopic titanium grooves with a width and depth of 200 μm in contact with osteoblast cells reveals a significantly higher value of ALP activity when compared to those without grooves. These results imply that grooves with greater width may accelerate the differentiation of osteoblast-related marker genes. However, this study was performed on titanium dental implants with the same width dimensions, and no data are available on the surface roughness of the disks tested [44]. A study on implant surfaces made of zirconia and textured with microgrooves reveals no statistically significant differences in ALP expression in microgroove groups compared to the control group and the sandblasted and acid-etched group [26]. However, they only rated the different spacings, and there is no information about the width of the microgrooves tested. 

Additionally, a statistically significant difference was found in osteopontin expression between group A and control. It is likely that the addition of microgroove patterns (127 μm in width) influenced the behavior of the osteoblast cells by increasing the secretion of bone-related matrix proteins such as osteopontin [45]. Rezaei et al. found similar results in their study on the biological and osseointegration capabilities of hierarchical mesoscale grooves, microscale valleys and nanoscale nodules. They show that creating a hierarchically roughened morphology on zirconia influenced the behavior of the osteoblast cells, as it increased the secretion of osteopontin compared to machined surfaces [46]. Despite the higher value of the osteopontin expression found on milled microgroove groups compared to the control group, there was no statistically significant effect between them.

The fibroblast expression markers evaluated in this study were collagen and IL-8. Although the expression levels of collagen I were comparable between the control group and the width-standardized milling microgroove-textured groups, the expression levels of IL-8 on microgroove zirconia implant surfaces were higher with no statistically significant difference. A recent study by Iglhaut et al. with fibroblast cells cultured on grooved implant surfaces shows an increase in IL-8 expression levels [47]. However, they did not show the width of the grooves or the surface roughness values. In this study, despite the higher IL-8 values in the width-standardized milling microgroove-textured groups, no significant effect of the width-standardized milling microgroove-textured group was observed.

These results suggest that adding microgrooves to zirconia implant surfaces can improve their biological behavior, especially affecting the biological response of osteoblast cells, which are the main cells of bone tissue. This may improve the long-term survival and success of zirconia dental implants, particularly in patients who have undergone extraoral bone grafting techniques, or patients with systemic conditions based on a potentially greater amount of formed bone tissue and improved tissue sealing [48,49,50].

This in vitro study, despite its limitations, evaluated the cellular behaviour of the two key cell lines for the long-term maintenance of dental implants, osteoblasts for the osseointegration process and fibroblasts for soft tissue adaptation. A detailed evaluation of the cellular behavior on the zirconia surfaces is crucial, especially with these controlled production techniques. After in vitro validation, pre-clinical in vivo studies in animal models are essential to know if the biomaterial has a good biological response to undergo in vivo clinical tests. For the future, our group is testing bacterial colonization on these surfaces to improve implant surfaces and current major problems of long-term implant failure.

## 5. Conclusions 

The addition of different dimensions of microgroove patterns by conventional milling on sandblasted and acid-etched zirconia implant surfaces does not seems to improve cellular viability of human osteoblasts and gingival fibroblasts. However, large microgrooves on these gold standard zirconia surfaces appear to affect the main osteoblast markers’ differentiation, such as ALP and osteopontin. The improvement of cellular behavior by microgroove surface manipulation could be a procedure in biomaterials manufacturing to improve their long-term survival. 

## Figures and Tables

**Figure 1 materials-15-02481-f001:**
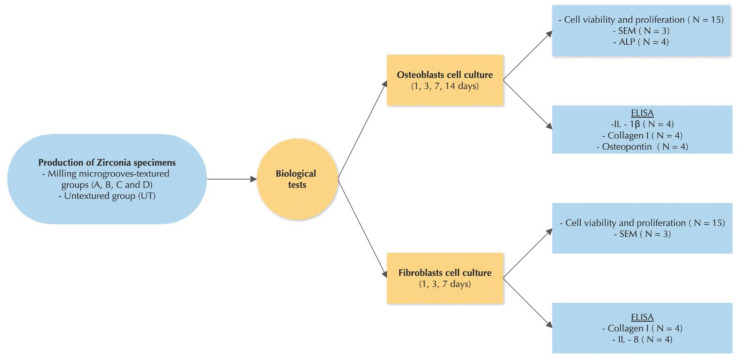
Study design detailing the methodologies used to develop this research work. The samples were divided in two groups and then biological tests were carried out, which was repeated in three independents experiments.

**Figure 2 materials-15-02481-f002:**
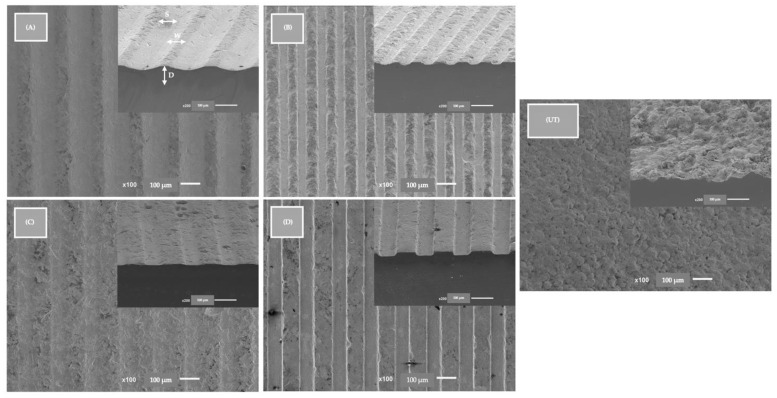
FEG-SEM images after blasting and acid etching of all study groups. Images at 100× magnification and cross-sectional images at 100× magnification at the same 90° angle. D—depth, W—width, S—spacing. Groups (**A**–**D**) were textured with width-standardized milling microgroove and control group (**UT**) were only sandblasting and acid etching. The scale bars are 100 μm.

**Figure 3 materials-15-02481-f003:**
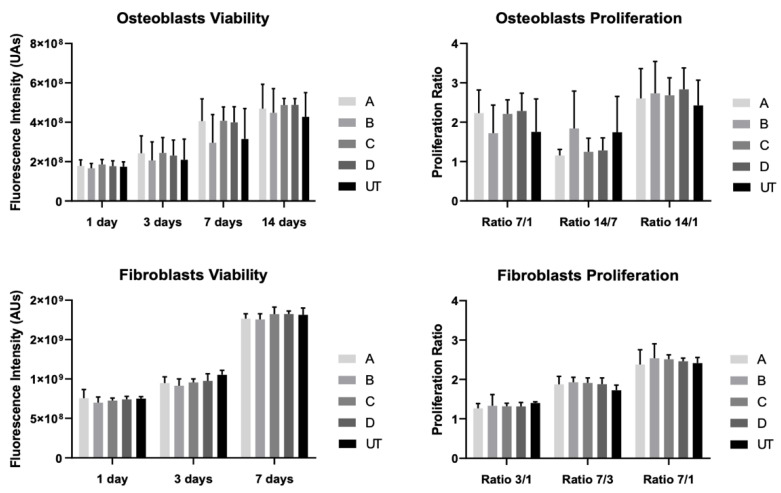
Fibroblast and osteoblast viability and proliferation measured as means and presented as bar graphs, with error bars representing the standard deviation. They were obtained in fluorescence intensity (AUs) and proliferation ratios after up to 14 days of culturing (N = 15). The group comparisons were based on one-way ANOVA with post-hoc Tukey.

**Figure 4 materials-15-02481-f004:**
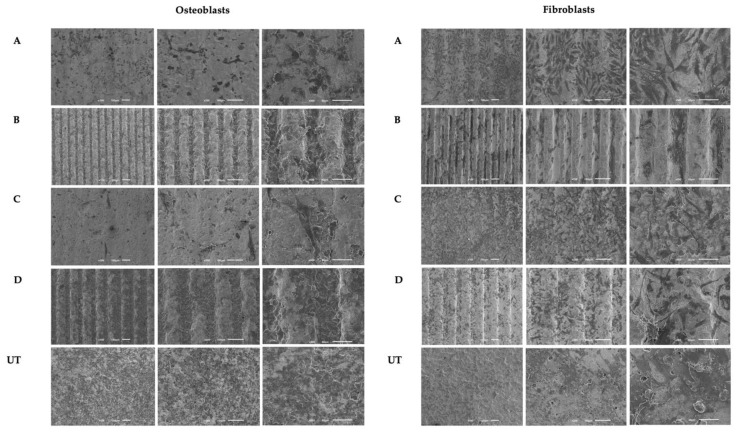
Images taken with FEG-SEM of fibroblast and osteoblast cultured on milled microgroove zirconia implant surfaces at 1 day (N = 3) with 100×, 200× and 500× magnification, respectively. The scale bars are 50 μm and 100 μm.

**Figure 5 materials-15-02481-f005:**
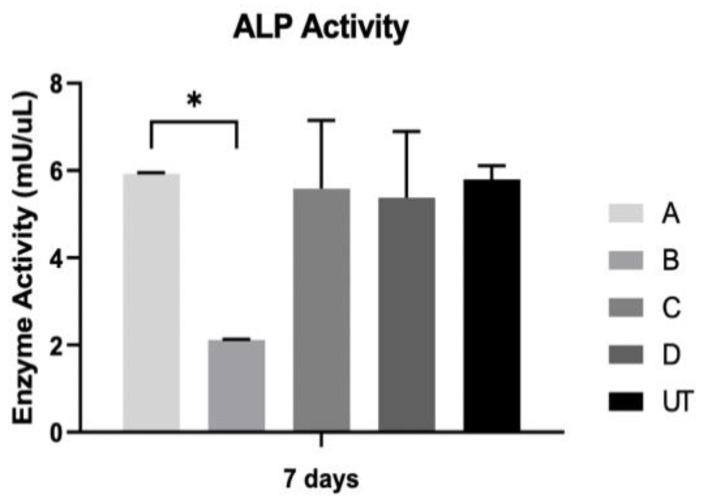
The activity of ALP was assessed on osteoblast cultures at 7 days (N = 4), presented in the form of bar graphs representing mean enzyme activity (mU/uL) and error bars representing the standard deviation. The group comparisons were based on Kruskall–Wallis with Dunn’s post-hoc tests. Level of significance: ∗ *p* < 0.05.

**Figure 6 materials-15-02481-f006:**
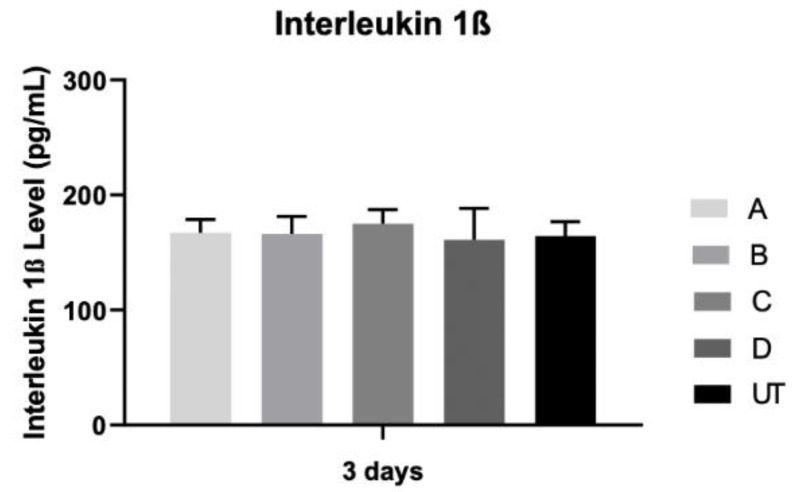
Interleukin 1β concentration (pg/mL) was assessed on osteoblast cultures at 3 days (N = 4), measured as means presented in the form of bar graphs, with error bars representing the standard deviation. Group comparisons were based on Kruskall–Wallis with Dunn’s post-hoc tests.

**Figure 7 materials-15-02481-f007:**
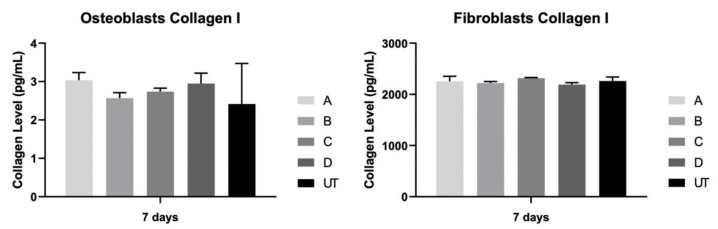
Concentration of collagen I (pg/mL) was assessed on fibroblast and osteoblast cultures at 7 days (N = 4), measured as means presented in the form of bar graphs, with error bars representing the standard deviation. The group comparisons were based on Kruskall–Wallis with Dunn’s post-hoc tests.

**Figure 8 materials-15-02481-f008:**
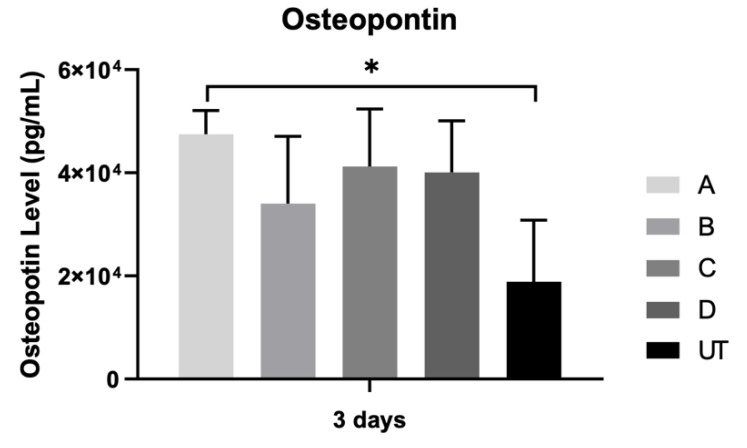
Bar charts showing mean concentration of osteopontin (pg/mL) measured at 3 days of osteoblast culturing (N = 4), measured as means presented in the form of bar graphs, with error bars representing the standard deviation. The groups comparisons were based on Kruskall–Wallis with Dunn’s post-hoc tests. Level of significance: ∗ *p* < 0.05.

**Figure 9 materials-15-02481-f009:**
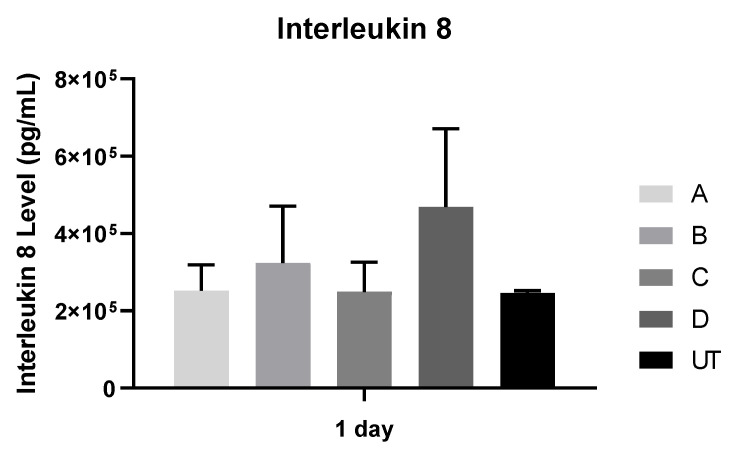
Mean concentration of Interleukin 8 (pg/mL) was assessed on fibroblast and at 1 day (N = 4), measured as means presented in the form of bar graphs, with error bars representing the standard deviation. The groups comparisons were based on Kruskall–Wallis with Dunn’s post-hoc tests.

**Table 1 materials-15-02481-t001:** Description of commercial 3Y-ZTP powder chemical compounds in accordance with manufacture Tosoh (Tosoh Corporation©, Amsterdam, The Netherlands).

Chemical Compounds	Y_2_O_3_	HfO_2_	Al_2_O_3_	SiO_2_	Fe_2_O_3_	Na_2_O
Weight percentage (wt%)	5.2 ± 0.5	<5.0	0.1~0.4	≤0.02	≤0.01	≤0.04

**Table 2 materials-15-02481-t002:** Schematic table of groove dimensions in μm obtained by SEM (N = 10).

Sample Group	Width (μm)	Depth (μm)
A	126.93 ± 3.76	12.26 ± 1.46
B	47.83 ± 2.05	10.42 ± 1.76
C	67.06 ± 2.04	8.41 ± 1.51
D	72.32 ± 2.77	17.07 ± 1.26

## Data Availability

The data presented in this study are available at: http://hdl.handle.net/10451/45607.

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
