# Peer review of "Human Gingival Fibroblast and Osteoblast Behavior on Groove-Milled Zirconia Implant Surfaces"

_materials, 2022, doi:10.3390/ma15072481_

Round 1

Reviewer 1 Report

This paper Human osteoblasts and gingival fibroblasts behavior on groove-milled Zirconia implant surfaces is an original study, with a total of 90 Zirconia discs randomly assigned by four width standardized milling microgroove-textured groups and control group without grooves. The authors have well approached the methods, working with osteoblasts and fibroblasts cultured on discs for a good timelapse (14 days). The introduction could be better focused on innovative similar studies (Bressan, E., Ferroni, L., Gardin, C., Bellin, G., Sbricoli, L., Sivolella, S., Brunello, G., Schwartz-Arad, D., Mijiritsky, E., Penarrocha, M., Penarrocha, D., Taccioli, C., Tatullo, M., Piattelli, A., & Zavan, B. (2019). Metal Nanoparticles Released from Dental Implant Surfaces: Potential Contribution to Chronic Inflammation and Peri-Implant Bone Loss. Materials (Basel, Switzerland), 12(12), 2036.).

The authors have measured also: Alkaline phosphatase activity, collagen I, osteopontin, interleukin 1b and interleukin 8 secretions. It is not clear why the authors have chosen different time-points to assess such markers: please explain. 

Conclusions should be increased with translational applications of such results.

Reviewer 2 Report

Interesting findings.

This paper deals with the response of osteoblasts and fibroblasts to various widths of microgroove-textured surfaces on 3Y-TZP zirconia discs that had been sandblasted. While the alkaline phosphatase activity and osteopontin secretion were significantly higher in the largest groove specimens, the viability of the cells was similar in all groups. However it was found that the deepest grooves had a positive effect on osteoblast differentiation, while there was no effect on the fibroblasts. Zirconia used for dental implants is in its infancy compared to those made from titanium and information on the role of surface characteristics is important especially given the high rates of failure seen in early implants made from this material.

As previously stated, there is a need to explore the response of the cells involved in osseointegration on zirconia surfaces. This well designed study found that the largest microgrooves created by milling had the greatest positive effect on osteoblasts. A major weakness of this study is that the cells used in this study have been shown to exhibit different behavior on the curved surfaces of implant bodies compared to that seen on discs. Also, the true test of these surface modifications should be explored using animal histology.

However, it seems appropriate to mention that cell behavior on discs may be different than on the curved surfaces of implants.

It was difficult to interpret Figures 1, 2, and 4.Reviewers need not comment on formatting issues that do not obscure the
meaning of the paper, as these will be addressed by editors.

Reviewer 3 Report

The study investigated the cellular response to the microgroove-milled zirconia surfaces. The topic is very interesting and has clinical relevance. However, I have a few comments as follow before the publication:

In the biological study, only four samples were used in parallel. There are no repeated tests. Considering the  biological variation, the biological test need to be repeated in a triple to avoid experimental errors

It is well-known that surface roughness influences cellular response. Please provide the data on surface roughness.

In Figure 4, please provide the scale bar.

Osteopontin plays a dominant role in biomineralization. Also, osteopontin is expressed in these mature cells. Please explain why the osteopontin concentration was measured after 3 days.

Round 2

Reviewer 3 Report

The manuscript can be considered for publication
